# Adolescent boys' sociocultural beliefs and attitudes toward menstruation in selected high schools in Ghana: Mediation and moderation effect of knowledge

Israel Wuresah[1]*, Priscilla Klutse[1], Sarah Odi Mantey[1], Isaiah Agorinya[1], Julie Hennegan[2], Sitsofe Gbogbo[1]

**1** Fred N. Binka School of Public Health, University of Health and Allied Sciences, Ho, Ghana, **2** Women's Children's and Adolescents' Health Program, Burnet Institute, Melbourne, Victoria, Australia

* iwuresah1996@gmail.com

## Abstract

Adolescent boys can reinforce negative societal attitudes towards menstruation and make it difficult for adolescent girls to achieve optimal menstrual health and hygiene (MHH) which defeats Sustainable Development Goals 3, 5.1, 5.2 and 5.6. This study investigated how knowledge mediates/moderates the association between sociocultural beliefs (SB) and attitudes of adolescent boys regarding menstruation in senior high schools in Ghana. A cross-sectional study was conducted in five mix-sex public senior high schools (SHS) in Volta region, Ghana. Probability proportional to size was used to distribute 431 study subjects among the schools, and surveys were conducted using structured questionnaires. Structural equation modeling (SEM) and generalized least square (GLS) modeling were conducted. The mean age of the sample was 17.3 (±1.0). Majority (60.6%) were between 15–17 years old, and Christians (95.4%). Half of them had low knowledge; 38% had moderate, and 11% had high knowledge. Meanwhile, negative SB (55.2%) outweighed positive SB, which reflected in their attitudes towards menstruation with 52.7% exhibiting poor attitudes. The SEM revealed that knowledge had a significant negative effect on SB and attitudes, with coefficients of -0.06 (SE = 0.01, p < 0.001), and -0.28 (SE = 0.06, p < 0.001) respectively. The GLS model indicated that knowledge also moderates the relationship between SB and attitudes. Male students in Ghana have a tendency to stigmatize menstruation among their female counterparts because they have inadequate knowledge about it and this may affect the achievement of optimal MHH by their female colleagues. It is evident that effective menstrual education reduces negative attitudes.

**Data availability statement:** The data supporting the findings of this study are available in the supporting information attached to this manuscript.

**Funding:** The authors received no specific funding for this work.

**Competing interests:** The authors have declared that no competing interests exist.

# 1. Introduction

Menstruation, an essential biological process, is often misunderstood and stigmatized in many cultures worldwide [1]. Such misconceptions lead to adverse societal attitudes, which significantly impact menstrual health and hygiene (MHH). These challenges extend beyond menstruating individuals, influencing broader societal dynamics, particularly in regions where cultural taboos are deeply entrenched. Among adolescent boys, these cultural beliefs and a lack of adequate knowledge reinforce stigma and discriminatory behaviors, making it more difficult for adolescent girls to achieve optimal MHH [2,3]. Several studies have been conducted on the knowledge, sociocultural beliefs, perceptions and attitudes of male adolescents regarding menstruation [2–6]. However, no known studies have exclusively attempted to indicate how knowledge mediates or moderates the impact of sociocultural beliefs on attitudes towards menstruation which is why this study is important.

It is public knowledge that traditions and cultural norms have been historically transmitted by men, which includes beliefs and practices associated with menstruation. Some of these practices, such as associating menstruation with impurity, are particularly harmful and contribute to systemic stigmatization [7]. During their formative years, adolescent boys often act as conduits for these societal attitudes, reinforcing existing norms and transmitting them to their peers and future generations [2,3]. This dynamic is further exacerbated by the lack of accurate knowledge about menstruation among boys. Khan et al. highlighted that many boys harbor significant misinformation about menstruation, which sustains and deepens the stigma surrounding it [8].

Despite a growing curiosity and a desire to understand menstruation better, boys frequently encounter barriers to gaining knowledge, such as restrictive educational frameworks and cultural silences maintained in homes and schools [9]. In Ghana, for example, studies reveal that boys' understanding of menstruation is frequently limited to superficial or erroneous information, which perpetuates harmful myths and negative attitudes [4,10]. This knowledge gap adversely impacts how boys perceive and interact with menstruating peers, deepening stigma and reinforcing gender inequalities.

The combination of cultural demands and inadequate education fosters negative attitudes among adolescent boys toward menstruation. These attitudes often manifest as teasing, avoidance, and a lack of empathy, creating barriers to constructive dialogues about menstruation. This does not only hinder boys' understanding of menstruation but also perpetuates gender inequities by limiting their ability to support menstruating peers. As Rana et al. noted, these dynamics exacerbate the challenges faced by menstruating individuals and contribute to a broader culture of silence and discrimination [6].

A key objective of this study was to model the role of knowledge as both a mediator and a moderator in the relationship between sociocultural beliefs and attitudes toward menstruation. The mediation hypothesis suggests that knowledge serves as a crucial explanatory mechanism through which sociocultural beliefs shape attitudes toward menstruation. Specifically, sociocultural beliefs, often rooted in misinformation and stigma, may directly contribute to negative attitudes, but the extent to which individuals acquire accurate knowledge may mediate this relationship by reducing biases

and misconceptions. As prior research has indicated, targeted education can reshape attitudes by correcting false beliefs and fostering more supportive perspectives [3,11]. Thus, it is hypothesized that higher levels of knowledge will weaken the negative effects of sociocultural beliefs on attitudes toward menstruation.

The moderation hypothesis posits that knowledge may also serve as a buffering factor that influences the strength of the relationship between sociocultural beliefs and attitudes. In this framework, adolescent boys who possess higher knowledge about menstruation may be more resistant to adopting stigmatizing attitudes, even when exposed to cultural norms that promote menstrual stigma. Conversely, those with lower knowledge levels may be more susceptible to the influence of such norms, reinforcing negative perceptions and discriminatory behaviors. This reflects the theoretical framework of cognitive buffering, which in this context implies that knowledge acquisition can mitigate the impact of harmful sociocultural norms (environmental influence) [12]. Fig 1 presents the concept of the study.

This study investigates how knowledge mediates and moderates the relationship between sociocultural beliefs and attitudes toward menstruation among adolescent boys in selected high schools in the Volta Region of Ghana. Adolescent boys are a group often overlooked in menstrual health education and research, thus, this study presents an opportunity to highlight the need to actively consider involving boys in the menstrual discourse for optimal menstrual health for females, improved overall health and gender equity. The implications of these findings are far-reaching, offering guidance for public health initiatives, educational reform, and policy development.

## 2. Methods and materials

### 2.1 Study design and setting

This study utilized a cross-sectional survey carried out across five districts in Ghana's Volta Region, which has a total population of 1.66 million, according to the 2021 Population and Housing Census [13]. The districts included in the study were Hohoe Municipality (population: 114,472), Afadzato South district (73,146), Kpando Municipality (58,552), Ho West District (82,886), and Ho Municipality (180,420). The Volta Region is predominantly inhabited by the Ewe ethnic group, with Christianity being the main religion.

### 2.2 Study population, sample size and sampling

Adolescent boys aged 15–19 years who were enrolled in selected senior high schools in Ghana and could understand and communicate in English were eligible to participate in the study. Also, inclusion required informed consent/assent from participants, with parental or guardian consent obtained where applicable.

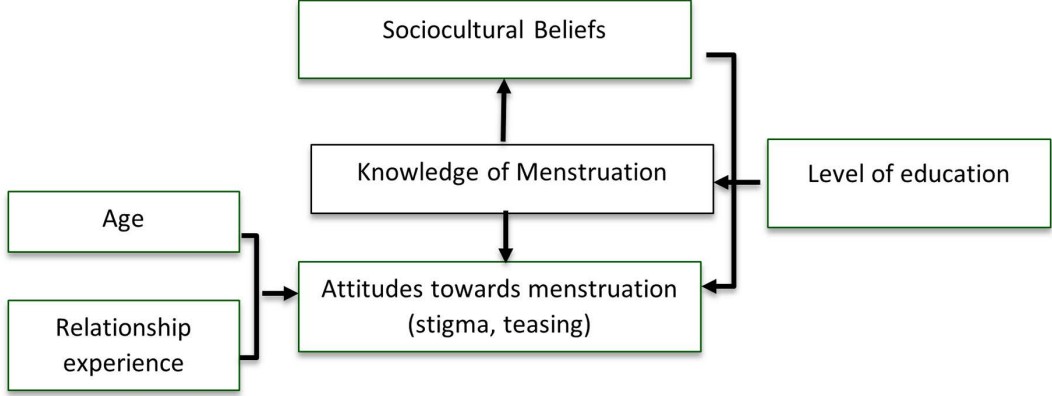

**Fig 1. Study's concept.**

At an assumed 50% prevalence of menstrual knowledge, with a 95% confidence level, 5% margin of error, 10% allowance for non-response, and a total population of 12,259 adolescent boys attending senior high schools across the five selected districts, 431 participants were included in this study using the Cochran's correction formula for finite populations [14]. This sample size was proportionally distributed among the selected schools based on enrollment figures.

The process of participant selection involved multiple steps to ensure randomness and fairness. Within each school, classrooms were chosen through a lottery system where class prefects picked from slips marked "Yes" or "No" where only those who drew "Yes" were included. A similar method was applied within selected classrooms to identify individual participants, with boys who selected slips marked "Yes" being enrolled. This approach ensured equal selection opportunities for all eligible students.

## 2.3 Data collection

A structured questionnaire, developed based on relevant literature, was used to collect data on participants' backgrounds, knowledge of menstruation and menstrual hygiene management, cultural or familial beliefs about menstruation, and attitudes towards menstruation. The tool, was digitized using Kobo Toolbox software and deployed on mobile devices (tablets and phones). Built-in system checks ensured data completeness before synchronization, enabling real-time error minimization and accuracy.

Data collection followed an interviewer-administered format conducted by trained male research assistants (RAs). After obtaining consent, one-on-one interviews were conducted in well-spaced environments within schools to maintain confidentiality and privacy. Participants were recruited and interviewed between April 28, 2024 and May 10, 2024. Collaboration with school authorities ensured the scheduling of interviews without disrupting academic activities.

At the end of each day, collected data were uploaded to a secure server under close supervision. This process facilitated efficient data management, timely error identification, and ensured data reliability and credibility.

## 2.4 Data management and analysis

**2.4.1 Measurement of variables.** This study examined the influence of sociocultural beliefs, knowledge of menstruation, and attitudes towards menstruation among adolescent boys. Sociocultural beliefs, the explanatory variable, were assessed using a 5-point Likert scale where composite scores were generated to represent the overall sociocultural beliefs. A lower mean score of the overall score indicated more positive sociocultural beliefs.

Knowledge of menstruation, acting as a mediator/moderator, measured boys' understanding of menstrual biology, hygiene, and related acuities. Knowledge scores, derived from correct responses to fact-based structured questions, were categorized into low, moderate, and high levels based on Bloom's cut-offs [15] for descriptive purposes, though treated as continuous variables in inferential analyses. Finally, attitudes towards menstruation, the dependent variable, were assessed through a 5-point Likert scale, capturing feelings, opinions, and perceptions of boys. A lower mean score indicated poor attitudes.

**2.4.2 Data analysis.** Descriptive analysis was performed to describe the cultural and family beliefs, knowledge, and attitudes prevalence against key demographic characteristics, juxtaposing for significant differences across these variables using the Chi square test. A statistically significant level of p-value <0.05 was applied.

Multivariate regression analysis using Structural Equation Modeling (SEM) was employed to investigate the mediating effect of knowledge about menstruation on the relationships between cultural beliefs and attitudes towards menstruation. The SEM analysis involved constructing a measurement model to capture the latent constructs of attitudes towards menstruation through their respective observed indicators while cultural beliefs and knowledge about menstruation were treated as observed constructs. Simultaneously, a structural model was formulated to explore and quantify the

hypothesized relationships among these variables. Iterative adjustments were made to enhance model fit. Estimations based on Sobel, Monte Carlo, Baron and Kenny [16] and Zhao et al., [17] were computed to infer mediation. The outcomes of the SEM analysis were carefully interpreted to provide understanding of how cultural beliefs and knowledge about menstruation contribute to shaping attitudes towards menstruation.

Also, a linear regression model with generalized least squares (GLS) estimation was ran to determine how knowledge about menstruation moderate the relationship between sociocultural beliefs and attitudes towards menstruation. The margins postestimation command was used to graphically display these interactions within the main variables. STATA v.17.0 was used for the statistical analysis.

## 2.5 Ethical considerations

Ethical approval was obtained from the University of Health and Allied Sciences Review Committee, along with permission from participating schools. Informed consent was secured from all participants, with parental assent obtained for those under 18. The study's purpose was clearly explained, and participants were informed of their right to withdraw at any time without consequences. Privacy and confidentiality were maintained throughout the study.

**2.5.1 Ethics statement.** The study was conducted per the Declaration of Helsinki and approved by the University of Health and Allied Sciences Research Ethics Committee UHAS-REC A.6 l3l23-24.

## 3. Results

### 3.1 Sociodemographic characteristics of participants

Table 1 presents the sociodemographic and household characteristics of adolescent schoolboys in Ghana's Volta region. Most students (60.6%) were aged 10–17 years, with Christianity being the predominant religion (95.4%), followed by Islam (3.7%). Ethnically, Ewes formed the majority (69.6%), with smaller representations from Akan (6.3%), Guan (4.9%), and Dagomba/Dagbani (1.9%) groups. Academic program distribution included General Arts (34.6%), Science (21.4%), Vocational (28.8%), and Business (15.3%), with more students in Form 2 (76.6%) than Form 1 (23.4%), reflecting Ghana's varying school calendars.

Parents were predominantly self-employed (49.9% fathers, 68.2% mothers), with some in public service (24.1% fathers, 15.1% mothers) or unemployed. Parental education levels ranged from no formal education (14.9% fathers, 9.7% mothers) to tertiary education (15.1% fathers, 4.8% mothers). Most students lived in compound houses (45.9%), followed by self-contained units (36.7%). Over half (55.9%) lived with both parents, while others resided with only mothers (22.5%), fathers (9.5%), or guardians (11.6%), and very few lived alone (0.5%).

### 3.2 Adolescent boys' knowledge, sociocultural beliefs and attitudes regarding menstruation

Fig 2 presents the descriptive statistics of the composite (main) variables of this study. Analysis using Bloom's cutoff [15] revealed that over half (51%) of adolescent boys exhibited low knowledge about menstruation, indicating significant gaps in understanding. A moderate level of knowledge was found in 38% of the boys, suggesting some awareness but also notable deficiencies. Only a small fraction, 11%, demonstrated a high level of knowledge, underscoring the overall inadequacy in menstrual health education among the group. A detailed description is attached as S1 Table.

The study further highlighted that negative sociocultural beliefs about menstruation were prevalent, with 55.2% of boys holding such views compared to 44.8% who expressed positive beliefs. Further data on boy's sociocultural beliefs is attached as S2 Table. Moreover, attitudes towards menstruation were predominantly poor among the boys, with 52.7% displaying negative attitudes. The data from which the scores were generated are presented as S3 Table.

**Table 1. Sociodemographic and household information of adolescent schoolboys in the Volta region.**

| Variable | Frequency, n (%) | | | | | χ² (p-value) | Total, n (%) |
|---|---|---|---|---|---|---|---|
| | Hohoe | Kpando | Ho West | Ho | Afadjato South | | |
| **Age (Mean, SD)** | 17.3 (± 1.0) | | | | | 2.5 (0.648) | |
| 15-17 | 29 (61.7) | 148 (62.5) | 48 (53.9) | 11 (68.8) | 25 (59.5) | | 261 (60.6) |
| 18-19 | 18 (38.3) | 89 (37.6) | 41 (46.1) | 5 (31.3) | 17 (40.5) | | 170 (39.4) |
| **Religion** | | | | | | 14.2 (0.076) | |
| Christian | 43 (91.5) | 228 (96.2) | 87 (97.8) | 16 (100) | 37 (88.1) | | 411 (95.4) |
| Islamic | 4 (8.5) | 7 (3.0) | 2 (2.3) | 0 | 3 (7.1) | | 16 (3.7) |
| Others | 0 | 2 (0.8) | 0 | 0 | 2 (4.8) | | 4 (0.9) |
| **Ethnicity** | | | | | | 47.1 (0.003) | |
| Akan | 5 (10.6) | 12 (5.1) | 3 (3.4) | 0 | 7 (16.7) | | 27 (6.3) |
| Dagomba/Dagbani | 2 (4.5) | 2 (0.8) | 1 (1.1) | 0 | 3 (7.1) | | 8 (1.9) |
| Ewe | 26 (55.3) | 166 (70.0) | 73 (82.0) | 14 (87.5) | 21 (50.0) | | 300 (69.6) |
| Ga/Dangbe | 2 (4.3) | 14(5.9) | 2 (2.3) | 2 (12.5) | 4 (9.5) | | 24 (5.6) |
| Guan | 2 (4.3) | 17 (7.2) | 2 (2.3) | 0 | 0 | | 21 (4.9) |
| Kokomba/Basare | 7 (14.9) | 21 (8.9) | 7 (7.9) | 0 | 5 (11.9) | | 40 (9. 3) |
| Others | 3 (6.4) | 5 (2.1) | 1 (1.1) | 0 | 2 (4.8) | | 11 (2.6) |
| **Study program** | | | | | | 94.3 (<0.001) | |
| Science | 5 (10.6) | 77 (32.5) | 1 (1.1) | 2 (12.5) | 7 (16.7) | | 92 (21.4) |
| Vocational programs | 24 (51.1) | 33 (13.9) | 37 (41.6) | 8 (50.0) | 22 (52.4) | | 124 (28.8) |
| Business | 3 (6.4) | 50 (21.1) | 12 (13.5) | 0 | 1 (2.4) | | 66 (15.3) |
| General Arts | 15 (31.9) | 77 (32.5) | 39 (43.8) | 6 (37.5) | 12 (28.6) | | 149 (34.6) |
| **Class** | | | | | | 410.6 (<0.001) | |
| Form 1 | 43 (91.5) | 0 | 0 | 16 (100) | 42 (100) | | 101 (23.4) |
| Form 2 | 4 (8.5) | 237 (100) | 89 (100) | 0 | 0 | | 330 (76.6) |
| **Male parent occupation** | | | | | | 39.0 (0.001) | |
| Unemployed | 6 (12.8) | 9 (3.8) | 5 (5.6) | 1 (6.3) | 2 (4.8) | | 23 (5.3) |
| Self- employed | 25 (53.2) | 102 (43.0) | 49 (55.1) | 11 (68.8) | 28 (66.7) | | 215 (49.9) |
| Civil Servant | 3 (6.4) | 75 (31.7) | 18 (20.2) | 1 (6.3) | 7 (16.7) | | 104 (24.1) |
| Private Employment | 5 (10.6) | 23 (9.7) | 13 (14.6) | 0 | 2 (4.8) | | 43 (10.0) |
| others | 8 (17.0) | 28 (11.8) | 4 (4.5) | 3 (18.8) | 3 (7.1) | | 46 (10.8) |
| **Female parent occupation** | | | | | | 38.0 (0.001) | |
| unemployed | 5 (10.6) | 11 (4.6) | 7 (7.9) | 0 | 3 (7.1) | | 26 (6.1) |
| Self -employed | 29 (61.7) | 149 (62.9) | 71 (79.8) | 13 (81.3) | 32 (76.2) | | 294 (68.2) |
| Civil servant | 1 (2. 1) | 42 (17.7) | 6 (6.7) | 1 (6.3) | 4 (9.5) | | 54 (12.5) |
| Private employment | 2 (4.3) | 14 (5.9) | 4 (4.5) | 0 | 1 (2.4) | | 21 (4.9) |
| others | 10 (21.3) | 21 (8.9) | 1 (1.1) | 2 (12.5) | 2 (2.4) | | 36 (8.4) |
| **Male parent education** | | | | | | 37.4 (<0.001) | |
| No formal education | 8 (17.0) | 17 (7.2) | 11 (12.4) | 0 | 6 (14.3) | | 42 (9.7) |
| Basic education | 20 (42.6) | 49 (20.7) | 29 (32.6) | 8 (50.0) | 13 (31.0) | | 119 (27.6) |
| Secondary education | 12 (25.5) | 79 (33.3) | 29 (32.6) | 6 (37.5) | 17 (40.5) | | 143 (33.2) |
| Tertiary education | 7 (14.9) | 92 (38.8) | 20 (22.5) | 2 (12.5) | 6 (14.3) | | 127 (29.5) |
| **Female parent education** | | | | | | | |
| No formal Education | 13 (27.7) | 30 (12.7) | 13 (14.6) | 2 (12.5) | 6 (14.3) | 30.7 (0.002) | 64 (14.9) |
| Basic Education | 25 (53.2) | 79 (33.3) | 41 (46.1) | 9 (56.3) | 18 (42.9) | | 172 (39.9) |
| Secondary Education | 7 (14.9) | 80 (33.8) | 23 (25.8) | 4 (25.0) | 16 (38.1) | | 130 (30.2) |
| Tertiary Education | 2 (4.3) | 48 (20.3) | 12 (13.5) | 1 (6.3) | 2 (4.8) | | 65 (15.1) |

*(Continued)*

**Table 1.** (Continued)

| Variable | Frequency, n (%) | | | | | | |
|---|---|---|---|---|---|---|---|
| | Hohoe | Kpando | Ho West | Ho | Afadjato South | χ² (p-value) | Total, n (%) |
| **Housing type** | | | | | | | |
| Compound house | 24 (51.1) | 92 (38.8) | 50 (56.2) | 8 (50.0) | 24 (57.1) | 16.6 (0.034) | 198 (45.9) |
| Private non-self-contained | 10 (21.3) | 48 (20.3) | 7 (7.9) | 4 (25.0) | 6 (14.3) | | 75 (17.4) |
| Self-contained | 13 (27.7) | 97 (40.9) | 32 (36.0) | 4 (25.0) | 12 (28.6) | | 158 (36.7) |
| **Live-in with** | | | | | | | |
| Alone | 1 (2.1) | 1 (0.4) | 0 | 0 | 0 | 34.4 (0.005) | 2 (0.5) |
| Both parents | 22 (46.8) | 149 (62.9) | 36 (40.5) | 12 (75.0) | 22 (52.4) | | 241 (55.9) |
| Father alone | 5 (10.6) | 19 (8.0) | 9 (10.1) | 0 | 8 (19.1) | | 41 (9.5) |
| Mother alone | 16 (34.0) | 42 (17.7) | 28 (31.5) | 1 (6.3) | 10 (23.8) | | 97 (22.5) |
| Guardian | 3 (6.4) | 26 (11.0) | 16 (18.0) | 3 (18.8) | 2 (4.8) | | 50 (11.6) |

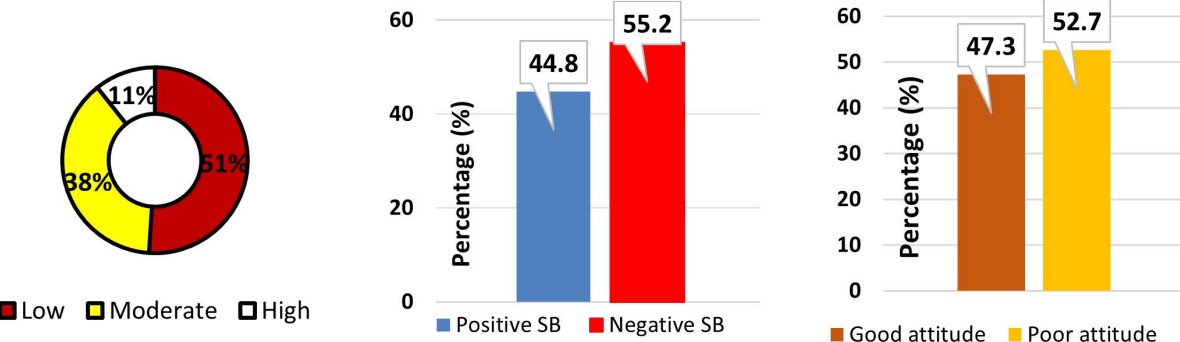

**Fig 2. Overall levels of knowledge, sociocultural beliefs and attitudes respectively.**

The figure provides the proportions of adolescent boys' knowledge on menstruation (left), which depicts that most of them had inadequate knowledge. Also, negative sociocultural beliefs (middle) outweighed the positives with observable traces to the attitudes (right) of adolescent schoolboys who generally exhibited poor attitudes.

The data presented in Table 2 reveals associations between sociodemographic characteristics and adolescent boys' knowledge, beliefs, and attitudes towards menstruation. Younger adolescent boys (15–17) exhibited more moderate to high levels of knowledge, while older boys (18–19) were more likely to have low knowledge. Ethnicity showed significant association with knowledge. This indicates that cultural background could be a factor in shaping perceptions, possibly reflecting differences in cultural narratives or educational access related to menstruation across ethnic groups. Academic focus significantly associates with knowledge levels, as students enrolled in science programs demonstrate higher knowledge, whereas those in vocational programs show lower knowledge. This disparity may stem from differences in curriculum emphasis, highlighting the importance of integrating menstrual health education across all academic disciplines to ensure broader awareness.

Class level was associated with attitudes as well, with Form 1 students showing more positive attitudes compared to their Form 2 counterparts. Further, boys with male parents who have no formal education exhibited poorer attitudes towards menstruation, while those with tertiary-educated male parents have better attitudes. Similarly, female parents' education levels significantly affect beliefs, with higher education correlating with more positive beliefs.

**Table 2. Associations with sociodemographic characteristics.**

| Variable | Menstrual knowledge | | | | Sociocultural beliefs | | | Attitudes | | |
|---|---|---|---|---|---|---|---|---|---|---|
| | Low | Moderate | High | $\chi^2$ | Negative | Positive | $\chi^2$ | Poor | Good | $\chi^2$ |
| **Age** | | | | | | | | | | |
| 15-17 | 120(46.0) | 109 (41.7) | 32(12.3) | 7.0* | 124(47.5) | 137(52.5) | 2.0 | 122 (46.7) | 139 (53.3) | 0.1 |
| 18-19 | 100(58.8) | 56 (32.9) | 14 (8.2) | | 69 (40.6) | 101(59.4) | | 82 (48.2) | 88 (51.8) | |
| **Religion** | | | | | | | | | | |
| Christian | 210(51.1) | 156 (38.0) | 45(10.9) | 3.3 | 185 (45.0) | 226 (55.0) | 2.7 | 197 (47.9) | 214 (52.1) | 1.5 |
| Islamic | 8 (50.0) | 8 (50.0) | 0 | | 5 (31.3) | 11 (68.8) | | 6 (37.5) | 10 (62.5) | |
| Others | 2 (50.0) | 1 (25.0) | 1 (25.0) | | 3 (75.0) | 1 (25.0) | | 1 (25.0) | 3 (75.0) | |
| **Ethnicity** | | | | | | | | | | |
| Akan | 17 (63.0) | 9 (33.3) | 1 (3.7) | 13.4 | 9 (33.3) | 18 (66.7) | 6.3 | 7 (25.9) | 20 (74.1) | 18.4* |
| Dagomba/Dagbani | 3 (37.5) | 3 (37.5) | 2 (25.0) | | 4 (50.0) | 4 (50.0) | | 4 (50.0) | 4 (50.0) | |
| Ewe | 148(49.3) | 119 (39.7) | 33(11.0) | | 139 (46.3) | 161 (53.7) | | 153 (51.0) | 147 (49.0) | |
| Ga/Dangbe | 13 (54.2) | 10 (41.7) | 1 (4.2) | | 10 (41.7) | 14 (58.3) | | 10 (41.7) | 14 (58.3) | |
| Guan | 10 (47.6) | 7 (33.3) | 4 (19.1) | | 13 (61.9) | 8 (38.1) | | 15 (71.4) | 6 (28.6) | |
| Kokomba/Basare | 25 (62.5) | 10 (25.0) | 5 (12.5) | | 14 (35.0) | 26 (65.0) | | 12 (30.0) | 28 (70.0) | |
| Others | 4 (36.4) | 7 (63.6) | 0 | | 4 (36.4) | 7 (63.6) | | 3 (27.3) | 8 (72.7) | |
| **Study Program** | | | | | | | | | | |
| Sciences | 35 (38.0) | 35 (38.0) | 22(23.9) | 29.5* | 49 (53.3) | 43 (46.7) | 6.3 | 54 (58.7) | 38 (41.3) | 10.4* |
| Vocational programs | 76 (61.3) | 43 (34.7) | 5 (4.0) | | 48 (38.7) | 76 (61.3) | | 46 (37.1) | 78 (62.9) | |
| Business | 32 (48.5) | 25 (37.9) | 9 (13.6) | | 25 (37.9) | 41 (62.1) | | 30 (45.5) | 36 (54.5) | |
| General arts | 77 (51.7) | 62 (41.6) | 10 (6.7) | | 71 (47.6) | 78 (52.4) | | 74 (49.7) | 75 (50.3) | |
| **Class** | | | | | | | | | | |
| Form 1 | 62 (61.4) | 33 (32.7) | 6 (5.9) | 6.6* | 42 (41.6) | 59 (58.4) | 0.5 | 35 (34.6) | 66 (65.4) | 8.5* |
| Form 2 | 158(47.9) | 132 (40.0) | 40(12.1) | | 151 (458) | 179 (54.2) | | 169 (51.2) | 161 (48.8) | |
| **Male parent educational level** | | | | | | | | | | |
| No formal education | 29 (69.1) | 10 (23.8) | 3 (7.1) | 14.7* | 14 (333) | 28 (66.7) | 5.9 | 11 (26.2) | 31 (73.8) | 19.1* |
| Basic education | 68 (57.1) | 37 (31.1) | 14(11.8) | | 51 (42.9) | 68 (57.1) | | 45 (37.8) | 74 (62.2) | |
| Secondary education | 68 (47.5) | 64 (44.8) | 11 (7.7) | | 61 (42.7) | 82 (57.3) | | 74 (51.7) | 69 (48.3) | |
| Tertiary education | 55 (43.3) | 54 (42.5) | 18(14.2) | | 67 (52.8) | 60 (47.2) | | 74 (58.3) | 53 (41.7) | |
| **Female parent educational level** | | | | | | | | | | |
| No formal education | 39 (60.9) | 19 (29.7) | 6 (9.4) | 11.1 | 22 (34.4) | 42 (65.) | 11.4* | 20 (31.2) | 44 (68.8) | 13.8* |
| Basic education | 83 (48.3) | 71 (41.3) | 18(10.5) | | 67 (39.0) | 105 (61.0) | | 78 (45.3) | 94 (54.7) | |
| Secondary education | 69 (53.0) | 52 (40.0) | 9 (6.9) | | 67 (51.5) | 63 (48.5) | | 65 (50.0) | 65 (50.0) | |
| Tertiary education | 29 (44.6) | 23 (35.4) | 13(20.0) | | 37 (56.9) | 28 (43.1) | | 41 (63.1) | 24 (36.9) | |

$\chi^{2*}$ = significant association at $p < 0.05$.

### 3.3 Mediation and moderation effects of knowledge on the association between sociocultural beliefs and attitudes towards menstruation

**Mediation.** The structural equation model (SEM) examined how knowledge mediates the link between sociocultural beliefs and attitudes towards menstruation among adolescent boys in Ghana's Volta region (see Fig 3). Results showed that increased knowledge reduces negative sociocultural beliefs (path coefficient: -0.06, $p < 0.001$) and negative attitudes (path coefficient: -0.28, $p < 0.001$). Sociocultural beliefs also contribute to negative attitudes (path coefficient: 0.11,

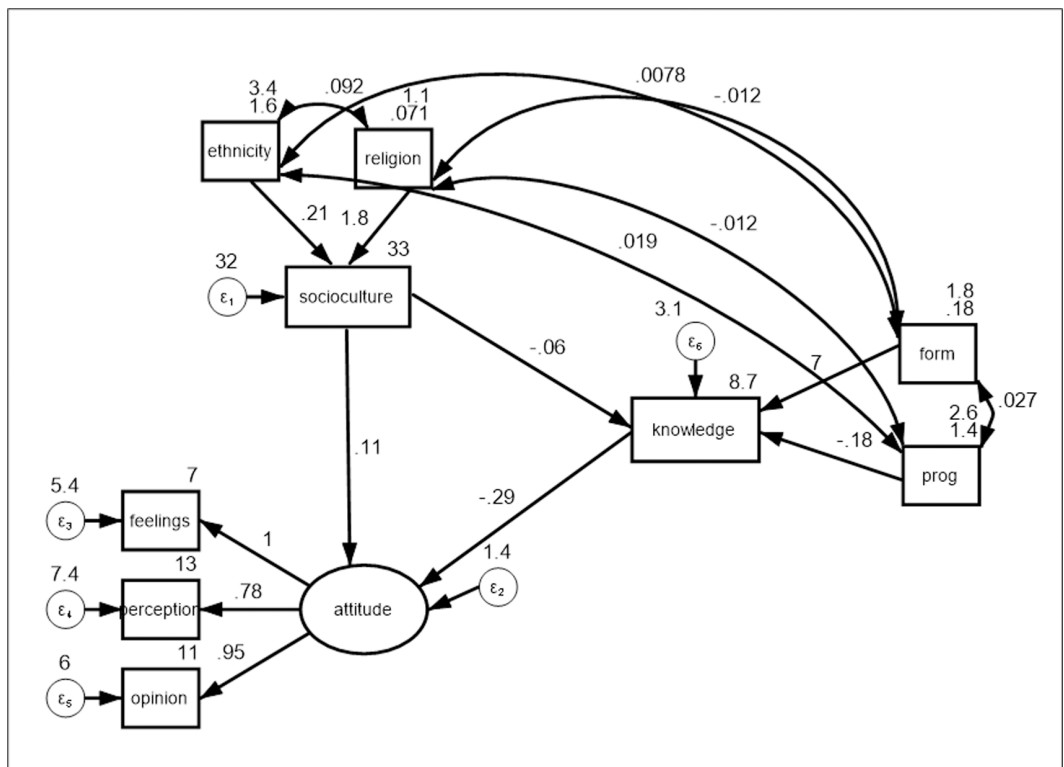

**Fig 3. Structural equation model of concepts.**

p<0.001) (see Table 3). Indicators of attitudes (perception and opinion) were robust measures. Model fit was reasonable, with RMSEA=0.06, CFI=0.9, but TLI=0.78 reflected sample limitations based on the arguments in literature [18–20].

The confirmatory analysis (Table 4) indicates that knowledge of menstruation partially mediates the relationship between sociocultural beliefs and attitudes towards menstruation. Indirect effects were significant (estimate: 0.052, SE: 0.015–0.016, p=0.001), as supported by Sobel and Monte Carlo tests. Following Baron and Kenny's framework, knowledge showed significant negative effects on sociocultural beliefs (B=-0.066, p<0.001) and attitudes (B=-0.787, p<0.001) while sociocultural beliefs positively influenced attitudes directly (B=0.309, p<0.001), also indicating partial mediation. Using the Zhao, Lynch, and Chen approach (2010), the mediation is classified as complementary (partial). The direct effect of sociocultural beliefs on attitudes remains significant, while the indirect path through knowledge contributes to the total effect. The Ratio of Indirect to Total effect (0.052/ 0.361) is 0.143, indicating 14% of the effect is mediated by knowledge. The Ratio of Indirect to Direct effect (0.052/ 0.309) is 0.167, indicating that the mediated effect is 17% the size of the direct effect.

**Moderation.** The Fig 4 illustrates moderation by showing how knowledge interacts with sociocultural beliefs to predict attitudes. The graph reveals two key slopes: $\beta1$=-0.018 (p<0.001) for lower sociocultural beliefs and $\beta2$=-0.045 (p<0.001) for higher sociocultural beliefs. The steeper slope associated with higher sociocultural beliefs suggests that increasing knowledge has a stronger negative impact on the predicted outcome for individuals with higher sociocultural beliefs. In contrast, when sociocultural beliefs are lower, the increase in knowledge results in a much smaller decrease in the linear prediction.

**Table 3. Estimation results of the structural equation model.**

| Path relationship | PC | SE | p-value | CI |
|---|---|---|---|---|
| **Structural** | | | | |
| Ethnicity ← Sociocultural beliefs | 0.21 | 0.22 | 0.352 | -0.23, 0.65 |
| Religion ← Sociocultural beliefs | 1.76 | 1.06 | 0.097 | -0.32, 3.85 |
| Sociocultural beliefs ← Knowledge | -0.06 | 0.01 | **<0.001** | -0.09, -0.3 |
| Program of study ← Knowledge | -0.18 | 0.07 | 0.015 | -0.32, -0.04 |
| Sociocultural beliefs ← Attitudes | 0. 11 | 0.02 | **<0.001** | 0.07, 0.15 |
| Knowledge ← Attitudes | -0.28 | 0.06 | **<0.001** | -0.4, -0.18 |
| **Measurement** | | | | |
| Feelings → Attitudes | * | | | |
| Perception → Attitudes | 0.78 | 0.16 | <0.001 | 0.47, 1.08 |
| Opinion → Attitudes | 0.95 | 0.16 | <0.001 | 0.64, 1.27 |
| | **Models' Goodness of fit estimates** | | | |
| $\chi^2$ | 235.89 | | <0.001 | |
| RMSEA | 0.06 (CI: 0.039 − 0.08) | | 0.199 | |
| CFI | 0.90 | | | |
| TLI | 0.78 | | | |

PC, path coefficient; SE, standard error; CI, confidence interval; *, constrained.

**Table 4. Confirmation of mediation using significance testing methods.**

| Test | Indirect effect | SE | p-value | 95% CI | |
|---|---|---|---|---|---|
| | | | | Lower bound | Upper bound |
| Sobel | 0.052 | 0.015 | 0.001 | 0.022 | 0.082 |
| Monte Carlo | 0.052 | 0.016 | 0.001 | 0.025 | 0.086 |
| **Baron and Kenny approach** | | | | | |
| **Mediation path** | **Coefficient (B)** | **p-value** | | | |
| Knowledge: Sociocultural beliefs (X→M) | -0.066 | <0.001 | | | |
| Attitudes: Knowledge (M→Y) | -0.787 | <0.001 | | | |
| Attitudes: Sociocultural beliefs (X→Y) | 0.309 | <0.001 | | | |

## 4. Discussion

The present study highlights significant gaps in adolescent boys' knowledge, beliefs, and attitudes toward menstruation, reflecting a broader societal neglect of menstrual health education for males. Over half of the boys exhibited low levels of knowledge about menstruation, with only 11% demonstrating a high understanding. This finding underlines systemic deficiencies in education systems that perpetuate menstrual health as a "female-only" topic, often leaving boys excluded from these critical conversations. The low levels of menstrual knowledge observed in this study mirror findings from similar research across other settings [3,5,21,22], which reinforces the notion that menstruation needs to be concealed and that males should not be involved in it. Meanwhile, the lack of exposure to menstrual education perpetuates taboos and myths, such as menstruation being a disease or a sign of impurity.

The higher knowledge levels observed among students in science programs suggest that curricula emphasizing bio-logical and health sciences play an important role in shaping awareness. Conversely, students in vocational programs showed significantly lower knowledge levels, indicating that unequal access to menstrual education across academic

 

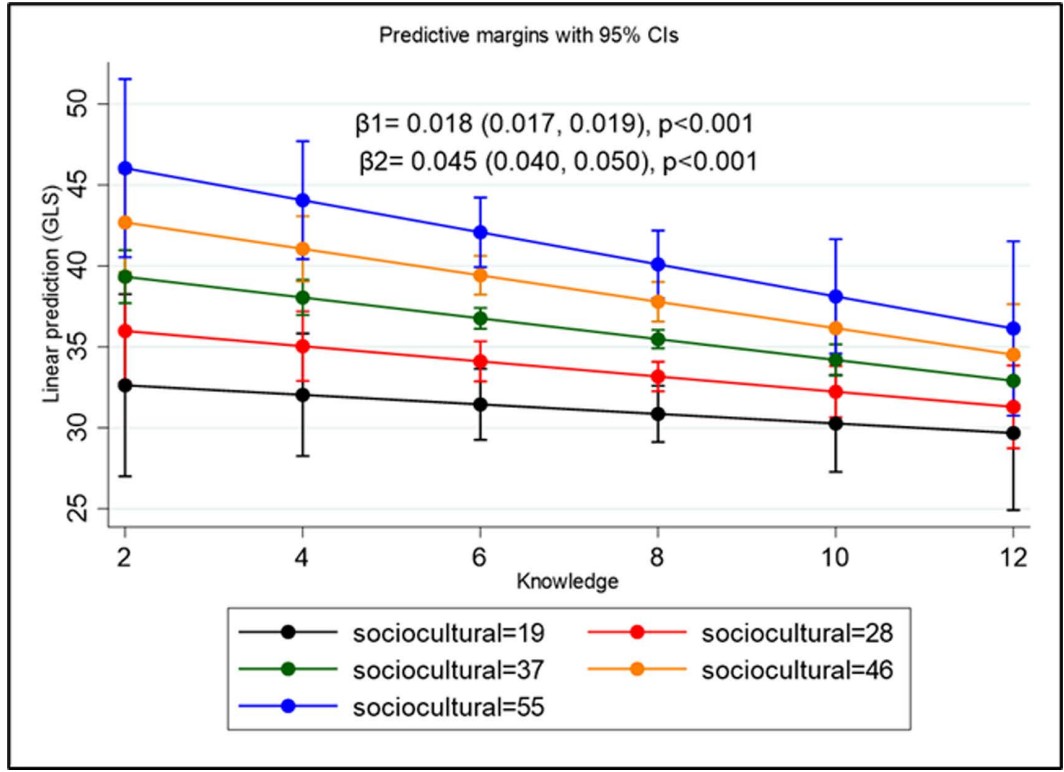

**Fig 4. Moderating effect of knowledge on the relationship between sociocultural beliefs and attitudes towards menstruation.**

disciplines limits awareness for many boys. Interestingly, the finding that younger boys (15–17) displayed higher knowledge levels than older boys (18–19) might reflect recent efforts in some basic schools to integrate menstrual health into earlier stages of education. This supports the work of Chandra-Mouli and Patel who emphasized that earlier exposure to menstrual education has a lasting positive impact on knowledge and attitudes [23]. However, the decline in knowledge among older boys highlights the need for sustained engagement across all educational levels to prevent attrition in understanding. Overall, these knowledge deficits perpetuate stigma, misinformation, and an inability to offer informed support to menstruating peers, thereby exacerbating gender inequities.

The study also revealed widespread negative sociocultural beliefs and poor attitudes toward menstruation among adolescent boys, with 55.2% holding negative beliefs and 52.7% exhibiting poor attitudes. Such findings highlight the deep-rooted cultural stigmas that frame menstruation as impure or shameful, further reinforcing taboos. These negative perceptions were more prevalent among boys whose parents, particularly fathers, had lower levels of education, suggesting a generational transmission of stigma. Ethnicity also indicates ties with boys' attitude, with cultural narratives varying significantly across groups. Boys from certain ethnic backgrounds were more likely to hold negative beliefs, indicating the role of community norms and cultural values in shaping perceptions.

Similar beliefs, where menstruation was viewed as a source of shame and secrecy among both boys and girls, were reported in a previous study. Such beliefs often discourage open discussions about menstruation and perpetuate misconceptions [3,4,22]. Further, the role of ethnicity in shaping menstrual beliefs and attitudes, as observed in the study, also finds parallels in existing research. A study by Coast et al. on menstrual health in sub-Saharan Africa emphasized that ethnicity and cultural norms are key determinants of menstrual perceptions and that communities with entrenched taboos are less likely to embrace positive attitudes [24]. These attitudes matter because they hinder the creation of inclusive

environments for menstruating individuals, perpetuate discriminatory behaviors, and undermine efforts to normalize menstruation as a natural process.

Additionally, the analysis showed that class level, and parental education significantly influence menstrual attitudes. Boys in Form 1 demonstrated more positive attitudes than those in Form 2, which could result from diminishing educational focus on menstrual health in later academic stages and also is indicative of some menstrual education efforts in basic schools. Parental education levels also proved critical, with boys whose parents, especially mothers, had tertiary education displaying more positive beliefs and attitudes. The association between parental education levels and more progressive attitudes, as highlighted in this study, supports findings by Crichton et al. who noted that parental influence significantly shapes adolescent attitudes toward reproductive health issues, particularly when parents themselves are well-educated [25]. These findings emphasize the importance of parental influence and intergenerational transmission of values, highlighting the need for community-wide education initiatives to dismantle entrenched taboos and promote positive attitudes toward menstruation.

The mediation and moderation analysis indicates that knowledge was shown to mediate and moderate the link between sociocultural beliefs and attitudes, with increased knowledge reducing negative beliefs and fostering more positive attitudes. However, the persistence of significant direct effects of sociocultural beliefs on attitudes suggests that knowledge alone cannot fully mitigate these negative influences. This highlights the intricate interaction of cognitive and cultural factors in shaping perceptions. Importantly, knowledge moderated the relationship between sociocultural beliefs and attitudes, with its impact being more pronounced among boys with higher levels of entrenched negative beliefs. This suggests that educational interventions may be most effective when targeted at individuals with deep-seated stigmas, as new information challenges and reshapes their preconceived notions. The partiality of the mediation aligns with theoretical frameworks like the Knowledge-Attitudes-Practices (KAP) model. This model posits that knowledge is a precursor to attitudinal and behavioral change, as supported by UNESCO's 2014 framework on comprehensive sexuality education [26]. The findings are further reinforced by Das et al. who demonstrated that education campaigns targeting menstrual knowledge significantly reduced negative attitudes among boys in India [27]. However, the partial mediation also suggests that while increasing knowledge reduces negative beliefs and attitudes, entrenched cultural stigmas remain a significant barrier. This finding parallels the work of Benshaul-Tolonen et al. who found that standalone education interventions often fail to fully address deep-seated sociocultural biases, underscoring the need for multifaceted approaches [28]. The moderation effect, where knowledge had a stronger impact on boys with more entrenched negative beliefs, supports cognitive dissonance theory, which suggests that exposure to new, evidence-based information can challenge and eventually shift rigid cultural narratives [29].

These findings have significant implications for policy and practice. Integrating menstrual health education across all academic disciplines, including vocational programs, is essential to ensure that no group of students is excluded. Further, community-level interventions are necessary to challenge the cultural stigmas surrounding menstruation, particularly in ethnic groups or regions with higher levels of negative beliefs. Parents, too, should be engaged in awareness programs, as their education levels and attitudes have a profound influence on shaping their children's perceptions.

### 4.1 Conclusion

In conclusion, the study reveals that adolescent boys' understanding of menstruation is shaped by multifaceted interrelations between knowledge, sociocultural beliefs, and attitudes, all of which are influenced by sociodemographic factors. Addressing these issues requires comprehensive educational strategies and culturally sensitive interventions to reduce stigma, improve knowledge, and foster positive attitudes. Such efforts will promote gender equity and contribute to a more inclusive and supportive society for all individuals.

### 4.2 Recommendations

Schools and public health organizations need to develop inclusive sexual and reproductive health curricula that address menstrual health and correct misconceptions for all genders, especially boys. There will be the need to involve parents

and guardians in these educational efforts given their significant role in shaping boy's knowledge. This will help reduce stigma and create a more supportive environment in schools and communities.

### 4.3 Strengths and limitations

Modelling knowledge as both a mediator and a moderator provide important data on the mechanisms underlying how conscious efforts to educate young people on menstruation can benefit the menstrual health of their female counterparts, in terms of yielding positive attitudes towards them. Thus, the study's findings have important implications for educational interventions and policy reforms aimed at fostering gender-equitable perspectives on menstruation.

However, the study also has some limitations. First, the study design has an inherent limitation where causal relationships between sociocultural beliefs, knowledge, and attitudes could not be studied. Secondly, both sociocultural beliefs and attitudes toward menstruation were self-reported by participants, which may lead to social desirability bias and a significant overlap between the constructs. This may affect the accuracy of the results, as participants may have provided responses that align with perceived social expectations rather than their true beliefs and attitudes. Lastly, we acknowledge that the relationship between knowledge and sociocultural beliefs can be bidirectional but this study did not explore this relationship. Future studies may consider this relationship.

### Supporting information

**S1 Table. General knowledge about menstruation among adolescent boys in the Volta Region.** This table presents responses to knowledge-based questions assessing the understanding of menstruation among adolescent boys (N = 431). (DOCX)

**S2 Table. Sociocultural beliefs about menstruation among adolescent boys in the Volta Region.** This table displays how participants (N = 431) perceive and experience cultural and familial beliefs surrounding menstruation. (DOCX)

**S3 Table. Attitudes towards menstruation and menstrual-related issues among adolescent boys.** This table presents statements on the attitudes of adolescent boys (N = 431) concerning menstruation. (DOCX)

**S1 Data. Dataset.** This file contains the dataset from which all analysis were conducted. (DTA)

### Acknowledgments

The authors acknowledge the support of the data collection team. We are also grateful to the teachers and administrators from all the Senior High Schools who assisted the research team during the data collection.

### Author contributions

**Conceptualization:** Israel Wuresah, Sitsofe Gbogbo.

**Formal analysis:** Israel Wuresah, Isaiah Agorinya.

**Methodology:** Israel Wuresah, Isaiah Agorinya, Sitsofe Gbogbo.

**Project administration:** Israel Wuresah, Priscilla Klutse, Sarah Odi Mantey.

**Supervision:** Isaiah Agorinya, Julie Hennegan, Sitsofe Gbogbo.

**Writing – original draft:** Israel Wuresah.

**Writing – review & editing:** Israel Wuresah, Priscilla Klutse, Sarah Odi Mantey, Isaiah Agorinya, Julie Hennegan, Sitsofe Gbogbo.

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
