## [Decision Letter · Decision Letter 0]

22 Apr 2025

PGPH-D-25-00348

Adolescent boys’ sociocultural beliefs and attitudes toward menstruation in selected high schools in Ghana: Mediation and moderation effect of knowledge

Dear Dr. Wuresah,

Thank you for submitting your manuscript to PLOS Global Public Health. After careful consideration, we feel that it has merit but does not fully meet PLOS Global Public Health’s publication criteria as it currently stands. Therefore, we invite you to submit a revised version of the manuscript that addresses the points raised during the review process.

hank for your interest in PLos GPH. One of our reviewer has asked for a minor comment. Kindly address them as it will add value to the paper. 

We look forward to receiving your revised manuscript.

Kind regards,

Muthusamy Sivakami

Academic Editor

Journal Requirements:

Additional Editor Comments (if provided):

This is a great paper.

Reviewers' comments:

Reviewer's Responses to Questions

**Comments to the Author**

1. Does this manuscript meet PLOS Global Public Health’s publication criteria?

Reviewer #1: Partly

Reviewer #2: Yes

2. Has the statistical analysis been performed appropriately and rigorously?

Reviewer #1: Yes

Reviewer #2: Yes

3. Have the authors made all data underlying the findings in their manuscript fully available (please refer to the Data Availability Statement at the start of the manuscript PDF file)?

Reviewer #1: Yes

Reviewer #2: Yes

4. Is the manuscript presented in an intelligible fashion and written in standard English?

Reviewer #1: Yes

Reviewer #2: Yes

Reviewer #1: I fund the manuscript well-prepared. However, I have the following comments to the authors.

1. You tried to show that knowledge has significant effects on sociocultural beliefs. How about the reverse? not explained (considered). There are conditions that socio-cultural beliefs can interfere scientific facts, knowledge and attitude. So, it is better to revise the conceptual framework of the study.

2. On the methodology part, it is better to describe the specific type of statistical test you used rather than describing by the type of model used. The sample size determination considered a prevalence of 50%, however there are previous reports of similar studies. The sample size using different variables also should have been extrapolated.

3. Inclusion and exclusion criteria are not mentioned.

4. The ethical issues, consents and privacy protocols for study participants are not clearly indicated. (mandatory)

Reviewer #2: The manuscript is technically sound and the data supported the conclusion that was drawn. The manuscript data however, weren't provided right away but the corresponding author has indicated that data will be made available on request. The authors have also provided supporting information at the end of the manuscript's pdf file.The manuscript therefore, meets the PLOS Global Public Health publication criteria.

**Do you want your identity to be public for this peer review?** For information about this choice, including consent withdrawal, please see our Privacy Policy

Reviewer #1: **Yes: ** Kumlachew Mergiaw Abtew

Reviewer #2: **Yes: ** Green, Pauline Aruoture

---

## [Editor Report · Decision Letter 1]

7 May 2025

Adolescent boys’ sociocultural beliefs and attitudes toward menstruation in selected high schools in Ghana: Mediation and moderation effect of knowledge

PGPH-D-25-00348R1

Dear Mr Wuresah,

We are pleased to inform you that your manuscript 'Adolescent boys’ sociocultural beliefs and attitudes toward menstruation in selected high schools in Ghana: Mediation and moderation effect of knowledge' has been provisionally accepted for publication in PLOS Global Public Health.

Best regards,

Muthusamy Sivakami

Academic Editor

The authors have addressed the comments raised by the reviews.